# Commons of the South: Ecologies of Interdependence in Local Territories of Chile

María Ignacia Ibarra [1,*], Aurelia Guasch [2], Jaime Ojeda [3,4], Wladimir Riquelme Maulen [5] and José Tomás Ibarra [4,6,7,*]

1 Department of Social Anthropology, Universitat de Barcelona, Catalunya, 08007 Barcelona, Spain
2 Faculty of History, Geography and Political Science, Pontificia Universidad Católica de Chile, Santiago 8331150, Chile
3 School of Environmental Studies, University of Victoria, Vancouver Island, BC V8P 5C2, Canada
4 Cape Horn International Center for Global Change Studies and Biocultural Conservation (CHIC), Universidad de Magallanes, Puerto Williams 6350001, Chile
5 Faculty of Architecture, Design and Urban Studies, Pontificia Universidad Católica de Chile, Santiago 8331150, Chile
6 ECOS (Ecosystem-Complexity-Society) Co-Laboratory, Center for Local Development (CEDEL) & Center for Intercultural and Indigenous Research (CIIR), Villarrica Campus, Pontificia Universidad Católica de Chile, Villarrica 4930445, Chile
7 Department of Ecosystems and Environment, Faculty of Agriculture and Forest Sciences & Center of Applied Ecology and Sustainability (CAPES), Pontificia Universidad Católica de Chile, Santiago 8331150, Chile
* Correspondence: mariaignaciaibarrae@gmail.com (M.I.I.); jtibarra@uc.cl (J.T.I.); Tel.: +34-631208112 (M.I.I.); +56-974528543 (J.T.I.)

**Abstract:** In a context of global social–ecological crises, a growing number of researchers, policymakers, activists and politicians have given importance to the "*commons*". This is mainly because the *commons* are associated with a logic of regulation and collective organization over the use and conservation of those goods considered essential for both human and ecosystem co-existence. This article seeks to draw attention to the *commons* from the standpoint of an ecology of interdependence and understand their modes of co-existence in the Global South. We analyze four case studies along with the tensions and junctures faced by the communities and the goods that sustain their continuity over time in southern Chile, a territory where extractivism and resource exploitation have increased over the last decades. The case studies use a combination of qualitative methodologies, including document analysis, literature review, ethnographies, participant observation, interviews and other means of participatory action research with community actors. Integrative analysis and discussion of the results reveal the fluidity and dynamism of the *commons* of southern Chile in contexts where there is pressure for their institutionalization and/or privatization, as well as various forms of resistance on the part of the territories for their protection and revitalization.

**Keywords:** extractivism; global south; Indigenous communities; neoliberalism; reciprocity; resistance

## 1. Introduction

Common goods, common pool resources, or the *commons* are concepts that have acquired ever more relevance in the context of multiple and interrelated social–ecological crises. In many cases, these concepts reflect the experiences of people in territories that organize themselves and, at the same time, face numerous crises in their daily lives. The *commons* have gained prominence in different currents of thought, principally because they are associated with a logic of regulation and cooperation for the sustainable use of those goods considered collective and essential for human existence and, often, also for that of the ecosystem. When thinking about common goods, it is necessary to bear in mind the contribution of Elinor Ostrom from economics [1,2], in showing that the tragedy of the *commons* [3] does not occur in many communities that sustainably and collectively

manage their resources based on the principles of trust and reciprocity [1]. Ostrom's research focuses on certain resources of common pool resources use (e.g., water basins, lakes, irrigation systems, fisheries, and forests) that remain sustainable despite being characterized by a high frequency of use, a high level of rivalry (which implies that they can only be used by a limited number of people at the same time), and great difficulty in excluding beneficiaries [2]. There are countless experiences that reveal the configuration of a micropolitics of resistance in defense of the praxis of the *commons* [4]. By redirecting our gaze towards the *commons*, we can learn about the experiences of pro-*commons* people and organizations [1,2] in the customary practices that have come to the fore in the current social–ecological crisis.

Laval and Dardot [5] proposed a conceptual shift from the category of common goods to that of the *commons*. The latter can be linked to a broader political use and claims that include everything that the communities consider fundamental for life and cannot, therefore, be privatized [5,6]. Thus, the *commons* are configured on the margins of capitalism, market economy, and top-down government decision making. They are constructed through community management, often carried out autonomously, in which rules that promote access and use by communities are recreated or recovered continuously [5,6]. This line of thought also includes the anti-capitalist approach proposed by, for example, Silvia Federici to understand the *commons*. She views them as autonomous spaces where there is shared property in the form of natural or social goods for the use of all the community members [7]. The *commons* are not only "things"; they are also expressed in imbricated ecological relationships of interdependence between human and nonhuman actors [8]. For this reason, an "ecology of interdependence" can help understanding the *commons* as active processes of sympoietic co-construction, in which networks of interactions and frictions for shared common pool resources become so complex, that the existence of community members may be considered as contingent on active engagement with the other in local territories [8–10]. For this reason, Federici highlights the need for the will to cooperate, debate, negotiate, and learn to manage conflicts and disagreements associated with the commons [7].

At the global level, the institutionalization of neoliberalism within the structures of political and social organization in the 1970s and 1980s favored economic approaches such as self-regulation of the market and the privatization of natural resources. Social welfare policies and the management of common goods ceased to be part of the institutional framework [10,11]. In Chile, this economic approach took form during the military dictatorship of Augusto Pinochet (1973–1990), with the collaboration of the articulators of neoliberal thought (e.g., Milton Friedman) [12]. Numerous aspects of social and material life were privatized, with long-standing effects on urban areas, rural territories and species of economic importance, as well as matters such as water use rights [13]. For example, in 1974, under Decree Law 701 for forestry development, subsidies were introduced for plantations of radiata pine (*Pinus radiata*) and blue gum (*Eucalyptus globulus*) in Indigenous traditional territories of southern Chile. This process contributed to the privatization of public land at low prices [14], a trend that continued during the transition to democracy. As a result of this process, which now dates back five decades, many natural resources in Chile's southern territories passed from community to individual (private) management [14].

Given the current crisis of the *commons* and the complexification of this concept now in vogue, it is necessary to understand what they mean for the people who inhabit local territories. It is also important to examine why and how people manage what they consider fundamental for their lives, opposing both the notion of private property and that of a natural resource "as that which, through reason, can be exploited for human development" [15,16]. Reflecting the underlying frictions, the *commons* are increasingly revitalized and vindicated as a plethora of political projects that emerge strongly from the Global South [17], as an epistemological and geopolitical space that challenges the hegemonic-modern civilizatory model [18]. In recent years in Chile, social–ecological fabrics have become highly politicized, particularly since the 2019 social uprising and

during the subsequent constituent process, which was truncated in 2022 by the rejection of the proposed Constitution. In an element that is not exempt from conflict, this Constitution would have recognized the existence of the *commons*. Today, we understand that the conversation has lost ground at the institutional level due to the country's sociopolitical circumstances. However, this does not imply that, at a number of local territories, there has ceased to be a need to protect, revitalize, and vindicate the community dynamics that safeguard and promote the *commons*.

Thus, in light of the current sociopolitical context, we believe there is a pressing need to draw attention to some perspectives from the South on the *commons* from the standpoint of an ecology of interdependence. This article seeks to make the *commons* visible and to understand their modes of coexistence in the Global South through the analysis of four case studies in southern Chile: Pinto Market in Temuco (La Araucanía Region); traditional seed exchanges in Wallmapu, comprising several municipal districts -with a focus on Villarrica (La Araucanía Region); waters of river Huenehue in Tralcapulli, Panguipulli municipal district (Los Ríos Region); and fishing corrals in channels and fjords of Patagonia (Los Lagos Region). We also examine the tensions and conjunctures faced by communities and the goods that sustain their continuity over time in a country where extractivism and resource exploitation are on the rise. We analyzed these four case studies in which, as a group of five researchers—two women and three men—from different disciplines (anthropology, ecology, agronomy, geography, and biology), we explored the theme of the *commons*. We were all born in the territory that today comprises the State of Chile and, from this place, are committed to the political resistance of the South and its first inhabitants. We recognize that, in their bodies of knowledge, practices, beliefs, and feelings, they have built wisdom that should be made visible and acknowledged more widely. To this end, we describe the vicissitudes of each case and the urgency of generating genuine dialogues with other epistemologies and ways of inhabiting these territories. The integrative analysis and discussion of the case studies reveal the fluidity and dynamism of the *commons* of southern Chile in contexts where there are pressures for their institutionalization and/or privatization, as well as diverse forms of resistance by the territories for their protection and revitalization.

## 2. Materials and Methods

This study brings together four cases that currently exist in different local territories of southern Chile, an ecoregion (i.e., a "natural area" that functions as a conservation unit on a continental and global scale [19]) that is located within the "Valdivian Temperate Rainforest". This is a global biodiversity hotspot that has experienced high rates of extractivism and resource exploitation over the last decades (e.g., large-scale projects for hydroelectricity, industrial fishing, salmon farming, agro-industry, mass tourism, and plantation forestry) and, therefore, faces multiple situations of social–ecological conflicts [20]. The following cases were analyzed (from north to south): Pinto Market (Temuco, La Araucanía Region), the exchange of traditional seeds (Multiple areas -focus on Villarrica, La Araucanía Region), waters of River Huenehue (Panguipulli, Los Ríos Region) and fishing corrals (Chiloé, Los Lagos Region) (Figure 1).

This article presents an interdisciplinary reflection (from anthropology, ecology, agronomy, geography and biology) which integrates different research methodologies. Each of the case studies focused on the analysis of practices of interdependence and communities' care for the *commons*. We also described how some barriers and conflicts could undermine the ecology of interdependence among peoples, species, and common places and practices. Through these four cases, we learned about local perspectives and tensions in the management of the *commons*. Indigenous Peoples and Local Communities (IPLC) bring a territorial perspective and a particular approach to environmental issues, with intertwined tensions that inherently arise from divergent rules, regulations and practices over the *commons*. The selection of each case study responds to the diversity of the *commons* involved, allowing us to understand the different experiences, approaches and social practices that can occur in

local territories of the global south. Each case study was examined using a combination of qualitative methodologies, including document analysis, literature review, ethnography, interviews and other means of participatory action research with community actors. In this way, with the information obtained and in order for the cases to dialogue with each other, we built five development points under a conceptual framework of the ecology of interdependence. The four cases were analyzed based on the tensions they have experienced historically and at present, and those elements that allow sustainability of the *commons* over time.

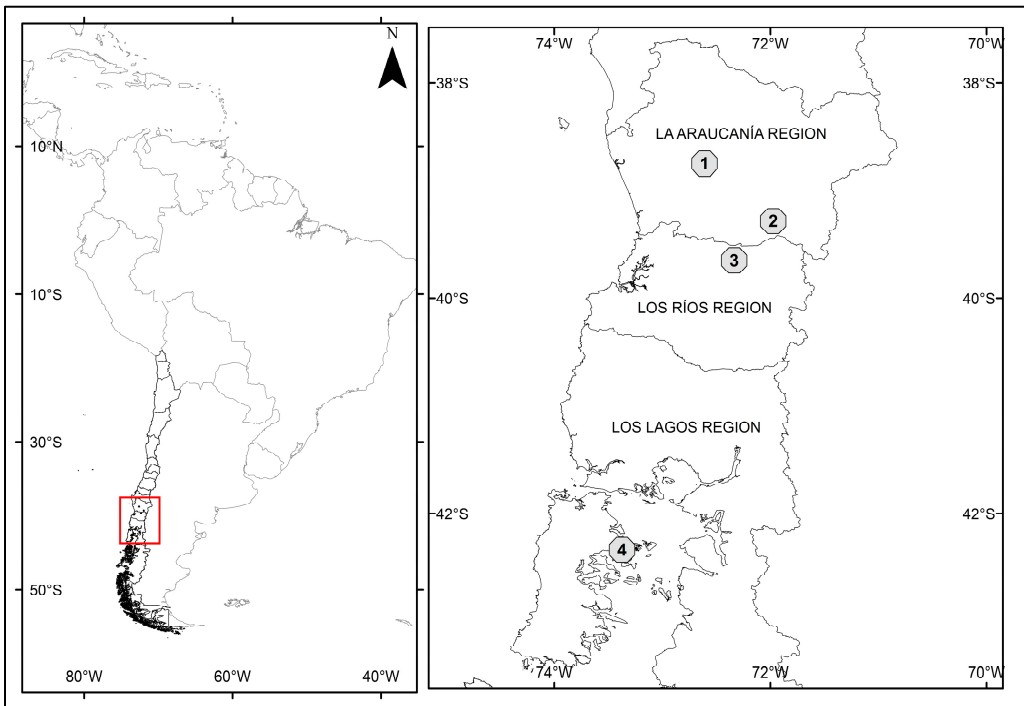

**Figure 1.** Spatial location of study cases of the *commons* in southern Chile: 1. Pinto Market (Temuco, La Araucanía Region), 2. Exchange of traditional seeds (Multiple areas—focus on Villarrica, La Araucanía Region), 3. Waters of River Huenehue (Panguipulli, Los Ríos Region) and 4. Fishing corrals (Chiloé, Los Lagos Region).

## 3. Results

Below, we briefly describe the main elements of the four case studies, using three categories of analysis: the *commons* in a context of ecologies of interdependence, emerging tensions and continuities of the *commons*. (Boxes 1, 2, 3 and 4).

**Box 1.** Case 1: Pinto Market.

> **Locality:** Temuco, Wallmapu, (La Araucanía Region, Chile).
> **Actors involved:** Stallholders in the Pinto Market.
> **Field methodology:**
> Ethnography is the methodological support, in terms of qualitative method and ethnographic approach to know the points of view and experiences of those who live social events. This was conducted through periodic visits to the Pinto market over four years, in which participant observation and semi-structured interviews with tenants and buyers were applied. The fieldwork process was complemented with a review of documentary sources on the Pinto market.

**Box 1.** *Cont.*

---

**The *commons* in local ecologies of interdependence:**

Street markets are a common space where agricultural and livestock products, groceries and other goods are exchanged (traded and bartered) by people who visit the market or frequent the streets adjacent to a central space administered by an institution. Street markets have been a feature of Chilean cities since the beginning of the 20th century and the Pinto Market was a pioneer in this field [21]. The Pinto market is still a small stronghold where rural and urban people come together, native seeds and vegetables, and culinary food traditions that evoke exchanges in good and cultural traditions, giving vitality to the *commons* as an interdependent network.

**Past and present emerging tensions:**

Tension in the Pinto Market is associated with the use of the common space, the different origins of the products exchanged there, and the relationship with institutions that intervene the market. It is a common space because it is (i) a historical urban space that has been used since the mid-1900s for the exchange of products of all kinds, (ii) a strategic space for relations at the regional and national level due to its strategic location in the city, and (iii) it is located next to the train station. At present, its consolidated sector in a large, covered area is institutionalized while the surrounding area is used by people who traditionally work as vegetable growers, algae gatherers, seed vendors, and used goods dealers or in other occupations of ordinary people. This creates tension about collective management, cleaning and the location of people in the market.

**Continuities of the *commons*:**

Street markets are spaces in which the local connects with global narratives from a local matrix. Given this premise, the Pinto Market has been connected to the global scale since its foundation: the train station was a transcendental milestone for its installation. However, in the current context where the local often succumbs to the increase in global interactions [22], street markets are spaces of the *commons* that protect and reflect what is local. Seeds that are part of global circulations become low cost compared to local seeds that continue due to the effort of family protection. The continuity of the *commons* in these spaces will depend on protecting this local and community matrix, so that the Pinto Market is a place where *the commons* become a physical space, human practices and goods that are exchanged on a daily basis.

---

**Box 2.** Case 2: Exchange of traditional seeds.

---

**Locality:** Wallmapu, numerous municipal districts, with focus on Villarrica (La Araucanía Region, Chile).

**Actors involved:** Mapuche Indigenous communities, organizations of mestizo campesinos, academics, consumer cooperatives, public agencies.

**Field methodology**:

Between 2016 and 2022, we have deployed several ecological surveys and ethnographic methods. These combine agrobiodiversity surveys in home gardens and local markets, participant observation, oral histories, interviews, free listing, food elicitation diaries, focus groups, and other means of participatory action research. All these methodologies have allowed us to get a broad understanding of, as well as a long-term commitment to, social–ecological fabrics associated with seed exchange and agroecological systems in Wallmapu.

**The *commons* in local ecologies of interdependence:**

In Wallmapu, the ancestral territory of the Mapuche people, Indigenous and non-Indigenous campesino women and men maintain agri-food practices that are rooted in biocultural memory. The latter is a web of knowledge, practices, beliefs and feelings that are present in a territory and are handed down from generation to generation. In home gardens and family farms, the *commons* are expressed in a myriad of traditional seeds (broad beans, other beans, peas, quinoa and Mapuche corn) whose cultivation transcends the economistic logic of "yield". The exchange of seeds, agricultural products and associated knowledge (called Txafkintü), which is anchored in relations of trust and reciprocity, is the expression of models of governance and a solidarity economy of the *commons* that moves freely in the territories. For example, people trust one another regarding the quality of the exchanged seed, its origin and the information associated with its cultivation requirements (e.g., soil, water, and light). These models show an active engagement of farmers in local territories, which is a critical means for strengthening the resilience of local food systems, despite the historical and contemporary pressures on family farming [23].

---

**Box 2.** *Cont*.

> **Past and present emerging tensions:**
> The Txafkintü or exchanges of seeds and associated knowledge, challenges the homogenizing and colonial approaches that have historically pressured farmers to reduce their range of activities and specialize in only one economic sector. Historically, some state programs—agricultural development agencies—have asked farmers that, in order to obtain subsidies, they must choose an activity such as raising animals, tending home gardens, or related activities including canning, weaving, beekeeping, gathering wild foods or wood carving. Farmer's specialization in a particular activity only reduces their adaptive capacity or resilience to crises (a health crisis, a disease in crops or livestock, drought, a volcanic eruption, or an economic crisis). Moreover, campesino and other civil organizations are increasingly concerned about the signing of treaties that facilitate international trade and, it is thought could, eventually foster seed privatization processes [23].
>
> **Continuities of the *commons*:**
> In Latin American territories, such as Wallmapu, biocultural memory has endured despite the numerous symbolic and material losses to which it has been subjected. Numerous, genuine alliances between farmers, academics, educators and civil society organizations are being formed to advocate the consecration of the right to food (fair, healthy and ecological) that sustains practices and relationships of exchange and reciprocity through care for life and the future of traditional seeds. Finally, although biocultural memory gives consistency to efforts to maintain mobile practices of exchange of the *commons* (seeds and knowledge), the integrating and most tangible agent of all these elements is a genuine "agroecology of the *commons*". Without this agroecology, which advocates access to free seeds in order to continue nurturing biocultural memory around food, all these foundations will lack the basis on which to build the society and life that the territories demand.

**Box 3.** Case 3: Waters of River Huenehue.

> **Locality:** Tralcapulli, Panguipulli municipal district (Los Ríos Region, Chile).
> **Actors involved:** Communities of the Tralcapulli and Llongahue, Asociación Leufu Wueney­wue, ENEL Greenpower.
> **Field methodology:**
> Ethnographic fieldwork was carried out during nine months divided into three periods between November 2018 and February 2021. The places of work were mainly the of Tralcapulli and Llongahue, in the district of Panguipulli, in various points of Wallmapu (Temuco, Villarrica and Valdivia) and in Santiago de Chile. Semi-structured and in-depth interviews were conducted with local interlocutors and focus group in which people participated from Tralcapulli.
>
> **The *commons* in local ecologies of interdependence:**
> On both sides of the Andes, the Mapuche people are organized around rivers [24]. For them, as a participant indicated in an interview: "the river is not only a river or only one; it is a network of connections of different bodies of water: the *menokos* (wetlands), the "taitas" or *trayenko* (waters that fall uniformly as springs that flow into the river), the *mallinko* (the marsh that then forms in a stream), lakes and glaciers". It is also streams that divide territories: the land grants given to the Mapuche after their defeat by the Chilean state were divided according to watercourses [25]. All the rivers in Wallmapu are considered ecological spaces that, as well as being a source of food (fish, herbs, freshwater mussels), have a cultural significance for the communities near them and are particularly important because of the spiritual relevance they each have [26]. People living in territories in resistance and in organizational experiences of conflict resolution have given way to the creation of strategies for the survival of the territory and social networks in contexts of crisis. There are multiple experiences, including conservation and recovery of seeds, community food, reforestation, and care economies.

**Box 3.** *Cont*.

> **Past and present emerging tensions:**
> Disturbance of the territory's physical geography has been reflected in profound changes in the systems of life that the Mapuche culture has associated with the River Huenehue. The channel has been raised and widened and water diverted from streams and springs for the benefit of the Pullinque hydroelectric plant and to the detriment of the population of Tralcapulli and Llongahue. Local voices point out that these changes and, above all, the closing of the sluice gates are the main causes of the disappearance of the river's ecological flow [27]. The River Huenehue is where the watercourse flowed before the Pullinque hydroelectric plant was installed on the riverbank. The community and the hydroelectric company, ENEL Greenpower, identify the river as a matter of dispute. The company considers it a natural resource while, for the community, it is a water landscape that encompasses a multitude of interdependent beings that determine each other mutually in a relational framework. From the neoliberal point of view, nature is not recognized as having an ontological imprint and relevance because it possesses a different logic from that of human beings [25]. The perspective that conceives them as utilitarian resources available to people to reproduce their anthropocentric system of needs has prevailed. There is a difference of perspectives, a mismatch between external intervention models and local practices [28]. The riverbank is part of a Mapuche geography composed of diverse fauna and plants used for local medicine, as well as being a sacred aquatic landscape [28]. In a neoliberal context, nature is not recognized as having an ontological imprint and importance because it possesses a different logic from that of human beings. The perspective that conceives nature as a pool of resources available to people to reproduce their anthropocentric system of needs has generally prevailed. Mapuche relational ontologies, on the contrary, often account for an interweaving of co-inhabitants who dwell in local places. This can be understood as that biodiversity or *itxofillmogen* that the Mapuche people seek to conserve, understanding the existence of multiple links between the beings that roam and reside on the land, understanding in turn the complex network and expanded diversity of life in the territory [27,28]. Hence, their relationship also with the protective spirits present in the various elements of nature, which they call ngen: "For us as Mapuche, nature itself has a spirit. The river has a ngen... Well, now that it is dry, I don't know if it is still there, alive. But that is more like, like when we, everyone says that 'the spirituality is lost' it is because the water is no longer running, the ngen is probably not there" [25]. Our ethnographic work conducted in southern Chile has shown that a good life or *küme mogen* requires reciprocity and interdependence between people and nature, in which ritualization and spirituality constitute a resistance mechanism to dispossession. On the banks of the Huenehue, there is constant tension between two paradigms: that of collaboration and recognition of an ecology of interdependence, and that of neoliberal individualism.
>
> **Continuities of the *commons*:**
> The cultural, economic and social importance of the river as a body of water is recognized in its role as a meeting space and in ecological relationships of interdependence between different species (human and nonhuman). In response, the community has sought its recovery and restoration of the waters, doing so in alliance with neighboring communities and local organizations to generate a lawsuit and challenge the State to recover the flow that existed in that place a few decades ago. This has, in turn, revitalized the community fabric, which has been continually threatened by extractive companies. The territory's local struggle is geared to the return of the body of water and its regeneration. A number of Mapuche self-determination practices reflect a Latin American version of political ecology: a deep articulation between nature and history [19] such as that seen in the recovery of the River Huenehue [27,28].

**Box 4.** Case 4: Fishing corrals.

> **Locality:** Channels and fjords of Patagonia (Los Lagos Region, Chile).
> **Actors involved:** Coastal Huilliche Indigenous communities, local Chiloé mestizo communities, artisan fishing people.
> **Field methodology:**
> We conducted a literature review using ethnographic and social–ecological references related to these marine innovations situated in the Patagonian channels (southern Chile). This search was carried out using keywords including "Corrales", "Fish Traps", or "Fishponds" through the Scielo and Google Scholar databases.

**Box 4.** *Cont.*

> **The *commons* in local ecologies of interdependence:**
> The coastal fishing corrals are structures made of stones building a rock wall. People built them in the intertidal zone, measuring about a meter high. They can be semi-circular, linear, or semi-rectangular and, when the tide rises, create an artificial pool where fish (e.g., *Eleginops maclovinus*) become captured [29]. Fishing corrals have been built by Indigenous peoples and mestizo-Chiloé communities and are found from the Llanquihue Province of the Los Lagos Region south to the Cape Horn archipelago [30]. Many communities and families living along the coast had management rights related to fishing corrals. The interdependence emerges when other animals use them; for example, coastal birds (herons) use the rock wall for fishing [29]. This network also appeared with common access and sharing habits. For instance, families invited neighbors to fish or gather shellfish in the corrals. This common access was also related to popular ceremonies or beliefs [29]. For example, it was believed that, if people do not practice generosity or fail to adhere to the ethical principles of sharing, a magical force and evil beings of nature could destroy the corrals and contaminate the beaches [30]. The traditional use of corrals has increasingly disappeared but, in some remote places (Caguach Island), these systems are still used for fishing or to store algae and shellfish.
>
> **Past and present emerging tensions:**
> This traditional practice has been degraded by cumulative ecological, social, economic, and political impacts. In 1960, the mega-earthquake in southern Chile (9.6 on the Richter scale) altered the elevation of the coast, changing the height of tides at various sites with fishing corrals. Then, in the mid-1970s and 1980s, the military dictatorship's neoliberal policies promoted industrial extractive fishing models. Many local communities recall how the abundance of fish dropped drastically due to fishing booms. For example, in the 80s, Patagonia faced the hake fishing boom—a fishery currently managed by individual transferable quotas (ITQs) [30]. In the 1990s, a salmon farming industry began to develop in southern Chile, expanding to many coastal areas with fishing corrals [29]. The pollution generated by this industry has had an important ecological impact. For instance, salmon farms can cause high levels of organic matter deposition, which reduces oxygen levels and species richness in the substrate [31]. Also, salmonids that have escaped from farms have the potential to disrupt trophic webs by eating native species like Odonthestes regia (fish related to fish ponds; 32). In addition, this industry can restrict marine access, altering the coastline's social configuration [32].
>
> **Continuities of the *commons*:**
> Indigenous marine governance initiatives have provided an opportunity for the continuity and revitalization of fishing corrals. Law N° 20.249, introduced in 2008 and known as the "Lafkenche Law", opened the way to the creation of Indigenous Peoples' Marine Coastal Spaces (ECMPOs). They seek to preserve Indigenous peoples' historical, sociocultural and ecological relations with the sea by granting them rights of access and protection of customary uses [33]. ECMPOs are a legal tool that can protect fishing corrals from industrial operations (salmon farming) and also serve as an opportunity for the cultural revitalization of these marine structures. In general, the threat of industrialization and privatization of the coastline has prompted local Indigenous and non-Indigenous communities to seek protection through ECMPOs or other conservation measures. For this reason, fishing corrals are being georeferenced and included in cartographies to reinforce their protection and revitalization.

Analysis of these four cases—the market, the traditional seeds, the river, and the fishing corrals (Figure 2)—reveals the web of pressures on the survival of the *commons* of the south. They can be classified into three types of tension: economic activities, public policies, and biophysical aspects of the territories where the *commons* are constituted.

The economic activities that exert pressure to use and benefit from *common* spaces and resources are, in general, extractivist and undertaken by Chilean or international companies, which are, in other words, exogenous to the territories. They include numerous energy projects, the salmon farm industry, industrial fishing, and initiatives to appropriate and eventually patent seeds. These activities seek to establish themselves in the territories, privatizing resources that have traditionally been used by the communities to sustain their existence. This process is precisely what David Harvey [34] refers to as *accumulation by dispossession* in explaining the dynamics through which local communities are deprived of access to natural or *common* resources, due to the arrival of companies and

capital that, supported by institutional and legal mechanisms, appropriate—or attempt to appropriate—them [35].

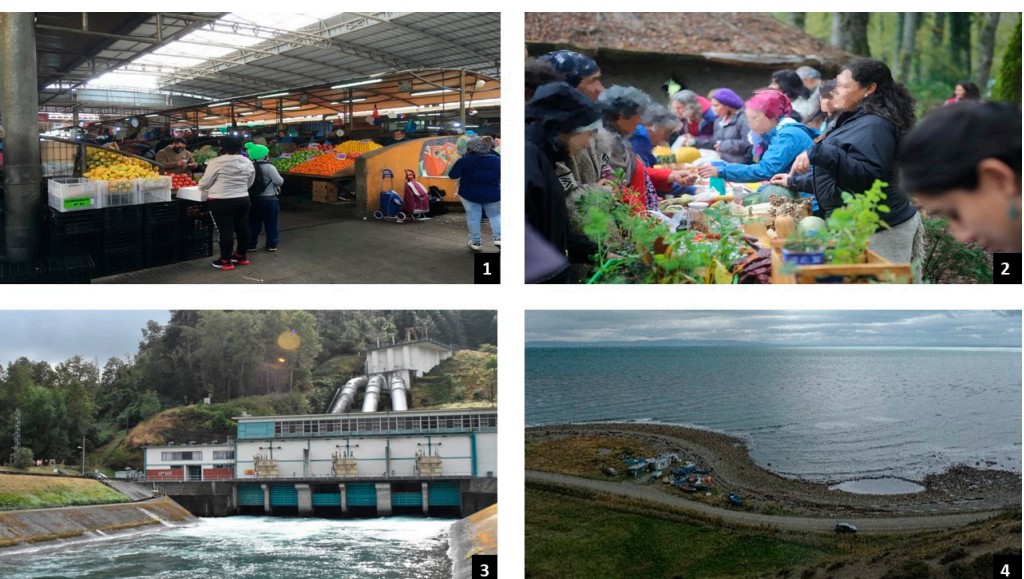

**Figure 2.** Scenes of study cases: (**1**) Pinto Market (Author: Wladimir Riquelme); (**2**) exchanges of traditional seeds and other agricultural products (Author: José Tomás Ibarra); (**3**) waters of the River Huenehue. (Author: María Ignacia Ibarra); (**4**) fishing corrals (Author: Alex García).

Although state public policies may be intended to "modernize" territories and "improve" their inhabitants' lives, they often dismantle or damage local logics and dynamics. This is seen, for example, in urban planning policies to regularize the space where the Pinto Market is located which, in their implementation, exclude its informal vendors, even though they have traditionally occupied this space to market their vegetables and other products. It is also seen in programs to help campesino communities modernize their agricultural production when they differentiate between those members of the community who can be beneficiaries and those who do not fulfill the required profile or simply do not wish to participate. In both cases, community practices are intervened and transformed based on a developmentalist imaginary that fails to take community ways of life into account.

Another of the tensions observed has to do with biophysical aspects of the constitution of ecologies of interdependence around the *commons*. One example of this is the 1960 earthquake which transformed the coastal geomorphology of southern Chile, affecting the traditional practices that occurred there. Thus, in social–ecological crises, there is a greater risk of alterations to these structures and, therefore, also the *commons*. Indeed, in the context of global climate change, some studies [36–38] have shown that it is rural, campesino and Indigenous communities that suffer the greatest impact despite making the least contribution to the phenomenon.

The integration of the four case studies also reveals three types of continuities of the *commons* of the South: the struggle to continue existing, the practices that give them life, and the state regulatory frameworks that sometimes ensure their maintenance. The struggle to exist is one of the main characteristics of the ecologies of interdependence of southern Chile. The communities that use and constitute the *commons* put up organized and conscious resistance, adopting with others interdependent strategies and alliances that contribute to maintaining access to the *commons* and their use over time. This demonstrates the vulnerability and fragility of the *commons* because they are configured on the margins of a capitalist system/world that absorbs and assimilates difference, transforming it into homogeneity [39]. In addition, it shows how the struggle itself is what sustains the continuity of the *commons*, even though they are under constant threat.

The ceremonies on the banks of the river Huenehue, the fishing corrals, traditional marketing by local producers in the Pinto Market, and the exchange of seeds and knowledge are practices conceived from and for the local community and are laden with reciprocity, biocultural memory, and the relationships of interdependence to carry them out. Relationships of reciprocity between people who inhabit the same space and struggle together for the commons account for bonds of collaboration, exchange, and interdependence. These are processes in which humans and nonhumans coexist and build a network of coexistence that mutually affect each other. These same community practices, their reproduction, and consistency are what allows the existence and continuity of the *commons* over time.

Finally, in some cases, legal frameworks guarantee the continuity of the *commons* of the South, protecting them against possible threats, tensions, and external pressures. This is the case of the Lafkenche Law (Law N° 20.249), which grants recognition to Indigenous communities' customary practices and uses of the coastal-marine zone. It should be noted that this was achieved after local communities worked together with social movements in a long negotiation process in the National Congress to address "ocean grabbing" [40] and safeguard their livelihoods.

## 4. Discussion

Analysis of the *commons* exemplified above in an urban market, seed exchange practices, an Andean river and coastal fishing corrals shows that systems of autonomous organization continue to be developed at the community level in different territories of southern Chile, despite the different pressures they face. This resilience of the *commons* (system's capacity to learn and endure by incorporating new information in response to broader social–ecological changes) is particularly important, because it emphasizes the sense of reciprocity between people and between people and nature [41]. For example, in defending the River Huenehue, Mapuche communities revitalize their systems of governance of the territory, reinforcing self-determination initiatives. At the same time, this defense of the river contributes ecologically to riverside biodiversity because it seeks to rebuild the environmental fabric in which the native fauna (birds, fish, and frogs, among other beings) has high value for the Mapuche families' food and productive activities [27]. Importantly, these social actions often have local ecological implications, but are rarely evaluated by researchers using inter- and/or transdisciplinary approaches. Therefore, we urge the creation of these links in order to gain an in-depth understanding of the inextricably linked social and ecological effects of local strategies of governance of the *commons*. Particularly in the current Chilean sociopolitical context, this discussion has implications given the process of drafting a new Constitution in which, due to historical privatization processes, the consecration of responsible administration of common goods is a source of controversy [42].

The results also reveal the fluidity and dynamism of the *commons* of southern Chile in withstanding pressure for top-down decision making under the public/private binary logic. They also enable us to understand the *commons* as a construction of ecological relationships of interdependence that are configured in frameworks of power based on agreements, negotiations, and strategies of resistance, which are created, after all, as a political construction of a collective project on a community scale [43]. In general, these strategies confront hegemonic social practices and gain some stability, recognition and protection over time only when translated into State regulatory frameworks.

We detected a gradient or hybridity of the *commons* in how they relate to and are structured in their situated ecologies of interdependence. Although all are managed communally, some accentuate a market logic under State norms as well as internal rules (for example, the market case) while others function autonomously through schemes and exchanges of a purely community nature (the river case and traditional seeds). Initiatives regarding the *commons* come to the fore during social, ecological and health crises such as COVID-19. In these contexts, the *commons* are expressed in initiatives such as community gardens, producer and consumer cooperatives, and seed exchanges that strengthen the

ecologies of interdependence. Thus, the results of our study allow us to understand how the existence of the *commons* contributes to community well-being, not only insofar as it guarantees access to and use of what is essential for existence, but also because it enables social organization and resilience in contexts of tensions and conflicts. The *commons* has the potential to reinforce feelings of capability and belonging as well as social cohesion. This, in turn, allows the communities that constitute them to look ahead to desired futures. In this context, the notion of ecologies of interdependence is an important concept for generating and repairing forms of collaboration to strengthen the *commons* because of its applicability in practice. This ecology of interdependence not only serves as a theoretical framework, but can also become a true "practice of transformation" that has, in the words of Escobar [18] (p. 99), the potential to be "a process of generating other worlds/practices, that is, radically changing the forms in which we encounter things and people, not just theorizing". This dynamic notion of interdependence could favor the generation of new ways of thinking about the complexities inherent in constantly evolving relationships [44]. In other words, this ecology of interdependence can serve as a practical tool for thinking relationally about how we "evolve" in relation to each other in contexts of rapid social–ecological changes that occur from the local to the global level [8].

The four case studies, which represent the *commons* of the South, have an Indigenous root which it is necessary to emphasize in the multiple reciprocal practices between the Mapuche people and their territory. The tragedy of the *commons* mentioned by Hardin [3] is contested from the territories [45], strengthening the theory that this position promotes the privatizing perspective. Derrick Jensen [46] affirms that this idea has been used by the political and economic right to dismiss the "common" use of the resources of the so-called "third world", where there are large Indigenous populations managing their resources. In general, this relationship is not one of individual property with an extractivist logic, but has to do with dialogue and the use of the elements in community terms. For example, the *commons* is translated into a traditional political-community organization, a system of norms that the *Kimche* (Mapuche wise people) call *Azmapu* (or Mapuche customary law) [47]. It consists in guidelines on behavior that are transmitted orally and regulate social relationships and ties with the land. In the river case, the plundering of its waters causes imbalances in Mapuche spirituality, directly harming families along its banks. The native vegetation that was born on the riverbanks has practical and spiritual significance for Mapuche medicine (*lawen*) and the threat has implications for community autonomy and the relational dimension between beings and plants that is expressed in the management of Indigenous communities' local healthcare [27]. *Kizugvnewkvlelafuy ta che* can be translated literally as "nobody is, or has been, their own boss" [48]. *Az Mapu*, or Mapuche law, also provides guidance on the relational dimension of the *commons* [15].

The Mapuche normative system conserves community dynamics of the ancestral Mapuche culture. However, given the territorial dispossession of which this Indigenous people has been a victim, there has been a destructuring of the roles that formerly led and/or administered power within local spaces. For example, the role of the *lonko* as community leader and organizational guidelines such as *Azmapu* have been systematically excluded under the colonial system established by the State of Chile and private economic interests. As a result, the system's regulation was relegated to dynamics of resistance within the communities, developing in constant tension and opposition to processes of co-optation by the State.

## 5. Conclusions

This study recognizes and highlights proposals in which the concepts of protection, solidarity, inter-subjectivity and dialogue are key for sustainability, visualizing the webs of diverse beings found in interdependent ecological fabrics that are not only *in* the world, but also *with* the world for the defense of life [21]. In the analysis of the study cases, the south emerges as a "subterfuge"; an "other" for each disposition imparted by the colonial matrix of power [49]. Human and nonhuman bodies are immersed in a world that forces

us to take into account the numerous other bodies that coexist in uncontrollable spaces far from self-sufficiency. The community fabric is composed of vulnerable and unfinished lives [50] that coexist in the territories and are constantly exposed to being affected by each other in dynamic and never-ending ecologies of interdependence. When we demonstrate that we live in an interdependent diversity, we recognize the meaning of plurality and community, of collaboration and mutual care as opposed to the individualistic vision. This relational notion of "interdependence" is similar to the Zapatista concept of the "pluriverse" (i.e., a world where many worlds fit), to acknowledge how ideas from non-Western worlds, in this case Indigenous resistance movements from Latin America, can shift the ways in which the *commons* are constructed [10,51]. Local community organization within the framework of ecologies of interdependence is urgently required to react to the oppression of the hegemonic capitalist machinery, amid the complexity of mutual contact that calls for a commitment to adaptation and reciprocity. The *commons* of the South project emphasizes the value of care of communities with their land that goes hand in hand with the territories' political and economic management.

**Author Contributions:** Conceptualization, M.I.I., A.G., J.O., W.R.M. and J.T.I.; methodology, M.I.I., A.G., J.O., W.R.M. and J.T.I.; formal analysis, M.I.I., A.G., J.O., W.R.M. and J.T.I.; investigation, M.I.I., A.G., J.O., W.R.M. and J.T.I.; writing—original draft preparation, M.I.I., A.G., J.O., W.R.M. and J.T.I.; writing—review and editing, M.I.I., A.G., J.O., W.R.M. and J.T.I. All authors have read and agreed to the published version of the manuscript.

**Funding:** J.T.I. acknowledges support from ANID/Fondecyt Regular (1200291, 1231664, and 1221057), the Center for Intercultural and Indigenous Research CIIR-ANID/FONDAP 15110006, the Center of Applied Ecology and Sustainability CAPES-ANID PIA/BASAL FB0002, and the Cape Horn International Center CHIC-ANID PIA/BASAL PFB210018. We thank Francisca Santana for preparing Figure 1. We thank the five anonymous reviewers for their careful reading of our manuscript and their many insightful comments and suggestions.

**Institutional Review Board Statement:** The study on traditional seeds was approved by the Ethics Committee of the Pontificia Universidad Católica de Chile (protocol code: 190603004; 24 April 2020).

**Informed Consent Statement:** Informed consent was obtained from all subjects involved in the study.

**Data Availability Statement:** Not applicable.

**Conflicts of Interest:** The authors declare no conflict of interest.

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
