# Peer review of "Commons of the South: Ecologies of Interdependence in Local Territories of Chile"

_sustainability, doi:10.3390/su151310515_

Round 1
Reviewer 1 Report
The paper can be published if it has a better approach on explaining the need of evaluation common goods as a trait for protecting natural resources or the environmental exploitation should be rationalised, or a deeper analysis of Elinor Ostrom's ideas on common goods and their characteristics from different point of view (the state, the exploration/exploitation companies and others). So, if the authors address this issue and also create in the Introduction section a paragraph dedicated to backbone and the methodological process behind the paper, I think it can be published.
Reviewer 2 Report
The manuscript entitled ‘Commons of the south: ecologies of interdependence in local territories of Chile’ by Ibarra et al. is a well written manuscript without any grammatical or typological errors. However, I do not feel the paper worth for publication. Below are the detailed comments;
a) The objectives of the study are not understandable;
b) Introduction is too lengthy; and has not justified the study clearly;
c) Hypothesis or clear objectives of the study are missing;
d) The methodology is not clear and thus not replicable;
e) In the result section; one can find discussion. Figures, tables as well as analysis is missing;
f) Overall the manuscript is not written in a scientific manner and is not replicable;
I can endorse the manuscript for the publication.
Reviewer 3 Report
Please see the attached review report.

Author Response
Dear Reviewer,
Please find the attachment.
kind regards

Reviewer 4 Report
# General Comments:
This study seeks to visibilize the commons from the standpoint of an ecology of interdependence and understand their modes of co-existence in the Global South. the authors analyze four case studies along with the tensions and junctures faced by the communities and the goods that sustain their continuity over time in southern Chile, a territory where extractivism and resource exploitation are increasing.
Overall, I think this manuscript is easy to follow and logically structured, the argumentation is convincing, is a solid piece of work and meaningful that has a good potential to be published. Below I list a few comments for your reference.
# 1.The biggest shortcoming of this manuscript is the lack of a theoretical analysis framework, or an analysis tool. If the author can find a suitable theoretical analysis framework to run through the whole manuscript, it will make the manuscript more theoretical and logical.
# 2. This manuscript mainly analyzes four cases: River Huenehue (Panguipulli, Los Ríos Region), fishing corrals (Chiloé, Los Lagos Region), Pinto Market (Temuco, La Araucanía Region), and the management of traditional seeds (Multiple areas, La Araucanía Region). Why these four cases and not the others? What are your selection criteria? What is the significance of choosing these four cases? It would make the article more meaningful if the author could give further explanation.
# 3. The author can make a brief introduction to the further research direction, so that more scholars will pay attention to and carry out the study of “Commons”.
# 4. The clarity of the figures in the manuscript is inadequate.
Reviewer 5 Report
Article review : Commons of the south: ecologies of interdependence in local territories of Chile
The authors described the purpose of the study as follows
„we believe there is a pressing need to visibilize some perspectives from the South on the commons from the standpoint of an ecology of interdependence. In this study, therefore, we examine the modes of existence of the commons of southern Chile, the tensions and junctures they face, and the elements that sustain their continuity over time. “
The construction of the article is correct , the authors referred to 42 bibliographic items in the article.
The authors in the article demonstrate the need for a common good that will enable the preservation of historical lifestyles of native peoples and areas.
The authors in a study of 4 cases describe the existing historical areas and lifestyles (behavior). Historical lifestyles are ecological, taking into account care for the environment. Cultural assets can be quickly lost if instruments are not introduced in the area to protect them and instruments to promote this lifestyle. Cultural heritage is considered a non-economic factor in land use and should be treated as a resource that, if properly managed, can be transformed into capital that can play an important role in sustainable development strategies. However, experience shows that preserving only historical conditions in a given area does not always bring satisfactory results. Sustainable development requires the reconciliation of many conditioned . The experience of post-communist countries shows that the common good is often treated as nobody's property, which means that there is no proper care for these areas. I think that the authors should also consider this thread in the article . There is certainly a need to protect cultural heritage.
I also recommend adding a map with the study areas.
Reviewer 6 Report
I consider the proposal of this paper interesting and original, but it would be necessary to proceed with some improvements. First, the bibliography is not sufficient and rigorous. You refer Laval and Dardot on page 2, second paragraph but in the references the book on commons does not appear. This bookk proposes an unusual reading of the common or the commons. You should cite and situate better to these authors. An additional lack in the text is the non discussion of economic concepts of common goods or resources. You could gain in clarity.
On page 3 what do you mean by eco-region?
I would improve the presentation of the four cases with some better description of traditional and former acctivities and the current transformation towards capitalist production systems.
There is no mention of the technological change in these places.
I would suggest a description for each case of the regress or progress of the communs and where or in what sector. Even if the result is mixed. What are the "objects" (technical and social) for each case.
Second, the four cases should be better analysed with a final consideration of the commonalities and differences of the four case studies.
The methodology is not well explained. How do you reach your conclusions? Who were interviewes? There is no issue of gender, for example?
In several paragraphs it lacks references to evidence, other documents than those already used?
A final revision of the style and English.
I consider that the discussion of the four cases is not sufficient. It lacks integreation. What is the common point. And the next question is related to that. Could you explain better the global South or South and the indiginous root?
Round 2
Reviewer 2 Report
The authors are not willing or able to revise the manuscript as per the suggestions. I am not satisfied with the author responses.
Author Response
First of all, we thank the editorial team of Sustainability Journal and the blind peers for each of their comments and observations for our article entitled Commons of the south: ecologies of interdependence in local territories of Chile. We have considered each of their comments and have sought to incorporate them into the modifications we have made to the article.
To reviewer number 2, he did not specify any comments. We are willing and ready to work on the document, but we cannot do it without his/her corrections in this round. Here we attach the first round response to Reviewer 2.

Reviewer 3 Report
please check the review report

Author Response
In the following tables we respond to the comments of revisor number 3, giving answers about the modifications we have made to the first version submitted to the journal.
We appreciate the opportunity to expand and improve this article.
We remain attentive to your response,
Kind regards,

Round 3
Author Response
We thank the editorial team of Sustainability Journal and the blind peers for each of their comments and observations for our article entitled Commons of the south: ecologies of interdependence in local territories of Chile.
We have considered the comment and made modification to the text. Mostly, because we agree with the appreciation that we could improve the evidence.
We have added: “In a neoliberal context, nature is not recognized as having an ontological imprint and importance because it possesses a different logic from that of human beings. The perspective that conceives nature as a pool of resources available to people to reproduce their anthropocentric system of needs has generally prevailed. Mapuche relational ontologies, on the contrary, often account for an interweaving of co-inhabitants who dwell in local places. This can be understood as that biodiversity or itxofillmogen that the Mapuche people seek to conserve, understanding the existence of multiple links between the beings that roam and reside on the land, understanding in turn the complex network and expanded diversity of life in the territory [27, 28]. Hence, their relationship also with the protective spirits present in the various elements of nature, which they call ngen: "For us as Mapuche, nature itself has a spirit. The river has a ngen... Well, now that it is dry, I don't know if it is still there, alive. But that is more like, like when we, everyone says that ‘the spirituality is lost’ it is because the water is no longer running, the ngen is probably not there" [25]. Our ethnographic work conducted in southern Chile has shown that a good life or küme mogen requires reciprocity and interdependence between people and nature, in which ritualization and spirituality constitute a resistance mechanism to dispossession.

Round 4
Reviewer 3 Report
All my comments have been addressed satisfactorily and I approve this manuscript.
Author Response
Thanks for your kind comments. Authors have made revision based on all comments.